# The critical importance of mask seals on respirator performance: An analytical and simulation approach

**Mingxin Xu**, **Peter Lee**, **David Collins***

Department of Biomedical Engineering, University of Melbourne, Melbourne, Victoria, Australia

* david.collins@unimelb.edu.au

**Data Availability Statement:** All relevant data are within the paper and its Supporting information files.

**Funding:** The author(s) received no specific funding for this work.

## Abstract

Filtering facepiece respirators (FFRs) and medical masks are widely used to reduce the inhalation exposure of airborne particulates and biohazardous aerosols. Their protective capacity largely depends on the fraction of these that are filtered from the incoming air volume. While the performance and physics of different filter materials have been the topic of intensive study, less well understood are the effects of mask sealing. To address this, we introduce an approach to calculate the influence of face-seal leakage on filtration ratio and fit factor based on an analytical model and a finite element method (FEM) model, both of which take into account time-dependent human respiration velocities. Using these, we calculate the filtration ratio and fit factor for a range of ventilation resistance values relevant to filter materials, 500–2500 Pa·s·m⁻¹, where the filtration ratio and fit factor are calculated as a function of the mask gap dimensions, with good agreement between analytical and numerical models. The results show that the filtration ratio and fit factor are decrease markedly with even small increases in gap area. We also calculate particle filtration rates for N95 FFRs with various ventilation resistances and two commercial FFRs exemplars. Taken together, this work underscores the critical importance of forming a tight seal around the face as a factor in mask performance, where our straightforward analytical model can be readily applied to obtain estimates of mask performance.

## Introduction

Filtering facepiece respirators (FFRs) and medical masks are widely used to reduce inhalation exposure of potentially harmful airborne particles. Medical masks and FFRs (collectively referred to here as masks) have also been recommended by the World Health Organization (WHO) as infection prevention and control measures by health care workers, including for protection against respiratory diseases such as COVID-19 [1], SARS, seasonal influenza, pandemic influenza and avian influenza [2].

The material and filtration efficiency of masks have been the focus of many studies, including examination of repeatability and comfort of masks made of melt-blown fabric and nanofibers [3, 4] and the study on filtration efficiency for different mask designs and non-

**Competing interests:** The authors have declared that no competing interests exist.

conventional mask materials [5–8]. However, face-seal mask leakage is also an important penetration route for aerosol particles [9–12], though far less examined. For example, mask classification testing standards [13–16] are implemented under the tight fit of the mask, whereas the face-seal gaps are excluded as test factors in these standards. However, factors associated with repetitive use [17, 18], non-compliant wearing [19], and change of body position [20] can enlarge gaps at the interface between the mask and the face (as shown in Fig 1), which will degrade the overall mask performance. For example, Chen et al. [21] conducted experiments on the penetration fraction of aerosols with different particle sizes through the face-seal gaps at a constant flow rate. Their results demonstrated that as the flow rate increases, the proportion of aerosol leaking through the face-seal leakage increases. Cho et al. [22] further studied the penetration rate of N95 masks, with experimental results indicating that the majority of particle penetration occurs through face-seal gaps. Rengasamy et al. [23] further studied the penetration rate of particles of different sizes in the presence of face-seal gaps, which indicated that the smaller particles have more inward leakage than larger particles in the presence of face seal gaps. In addition, fugitive flow through gaps in N95 and P100 masks [24] and the influence of the contact pressure on the size of face-seal gaps [25] have also been examined.

In short, while previous studies examining mask gaps have been primarily experimental, often for specific mask classifications/mask models/face-seal gap dimensions, an analytical examination would provide a more universal understanding of the importance of mask sealing for a range of mask types, materials and dimensions. In this work we examine the fundamental physics driving the reduction in filtration ratio as a function of mask gaps and filter material permeabilities. While inertial and adhesion forces will dictate particle sedimentation even in the presence of gaps, this being a complex relationship between particle size, charge and a mask's geometry, we seek to examine the airflow itself to give a conservative basis for understanding reductions in overall filtration ratio. In doing so we develop an analytical model

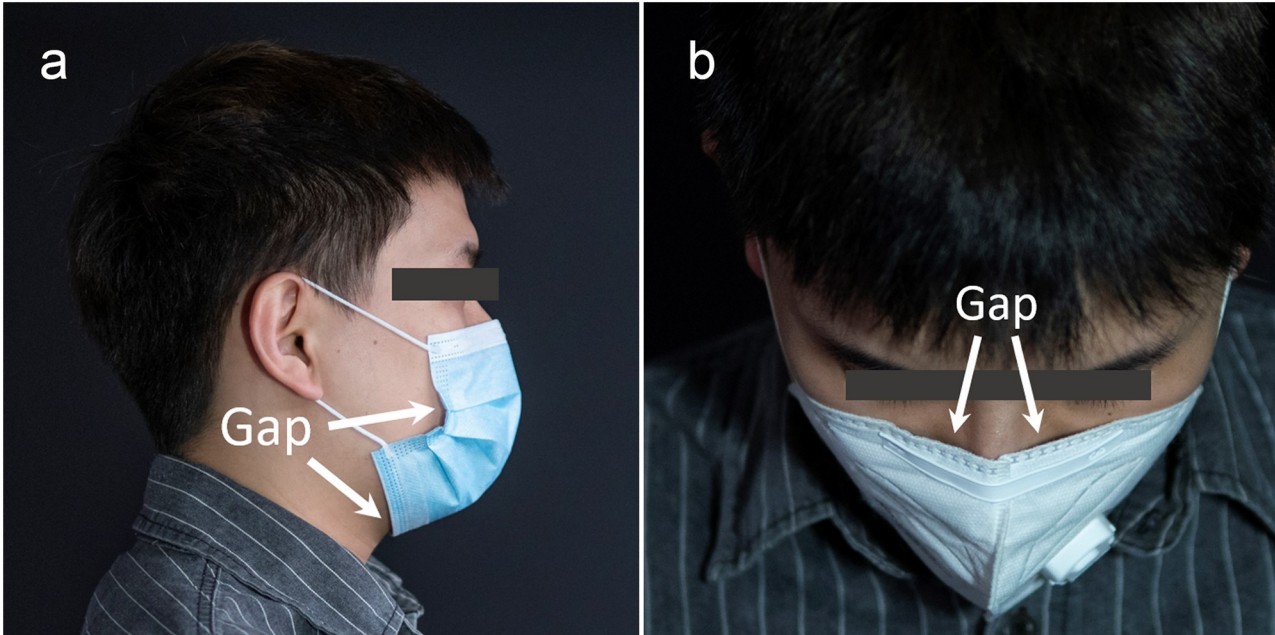

**Fig 1. Examples of face-seal gaps.** For (a) surgical mask and (b) N95 mask, and arrows indicating in (a) buccal gaps and (b) nasal gaps. The individual in this manuscript has given written informed consent (as outlined in PLOS consent form) to publish these case details.

incorporating time-dependent breathing rates representing an adult male and female, and corroborated using finite element model (FEM) simulations. By having a clear understanding of the critical relationship between mask gaps and the fraction of incoming air that enters via the filter material, we can provide first-order estimates of filtration ratio across a wide range of potential mask materials and gap dimensions, the latter being a function of material compliance, geometry and user factors. Our results highlight the critical importance of forming a good seal around the mask edges, especially true for lower-permeability (higher pressure-drop) filter materials. The results of this study will enable both mask designers and manufacturers to provide conservative estimates of filtration ratio, while reinforcing the importance of good mask fit for users.

## Models

### Respiration model

The primary purpose of this work is to quantify the air flow fraction that bypasses the filter material, requiring a physiologically relevant measure of human breath flow. Since the flow rate varies with time over a breath cycle, we adapt a time-dependent respiratory model to use as the velocity boundary conditions of the inlet /outlet (i.e. the mouth and nostrils). The respiratory flow rate, $Q_x$, is expressed as [26]

$$Q_x = \frac{\alpha_x sin \ (\beta_x t)}{1000},\tag{1}$$

where the subscript x can be replaced by *in* or *out*, representing the inhalation and exhalation, respectively. The parameters $\alpha_x$ and $\beta_x$ can be obtained from Eqs 2 and 3:

$$\alpha_x = \frac{1000\beta_x V_M(f_{in} + f_{out})}{4f_{in}f_{out}}\tag{2}$$

$$\beta_x = \frac{\pi}{30}f_x\tag{3}$$

The parameters and their expressions used to calculate $\alpha_x$ and $\beta_x$ are given in Table 1.

### Governing equations of the mask and the gap

A representative conceptual model of the mask with a gap is shown in Fig 2. The boundaries include the mask material, the gap between mask and face, the mouth, and the face. Intuitively, an input in air flow from the mouth increases the pressure differential between the interior and exterior of the mask ($P_1 > P_2$) and drives air flow through both the semi-permeable filter

**Table 1. Parameters and expressions used to calculate $\alpha_x$ and $\beta_x$.**

| Parameter | Symbol | Expression | |
|---|---|---|---|
| | | **Male** | **Female** |
| Breath volume per minute [26] | $V_M$ | $0.005225A_{body}$ $(m^3)$ | $0.004634A_{body}$ $(m^3)$ |
| Inspiratory frequency [26] | $f_{in}$ | $55.55 - 32.86H + 0.2602W$ $(Hz)$ | $46.43 - 18.85H$ $(Hz)$ |
| Expiratory frequency [26] | $f_{out}$ | $77.03 - 45.42H + 0.2373W$ $(Hz)$ | $54.47 - 25.48H$ $(Hz)$ |
| Estimated body surface area [27] | $A_{body}$ | $0.0235 \cdot (100H)^{0.42246} \cdot 100W^{0.51456}$ $(m^2)$ | $0.0235 \cdot (100H)^{0.42246} \cdot 100W^{0.51456}$ $(m^2)$ |
| Average adult body height in Australia [28] | $H$ | $1.756$ $(m)$ | $1.618$ $(m)$ |
| Average adult body weight in Australia [28] | $W$ | $85.9$ $(kg)$ | $71.1$ $(kg)$ |

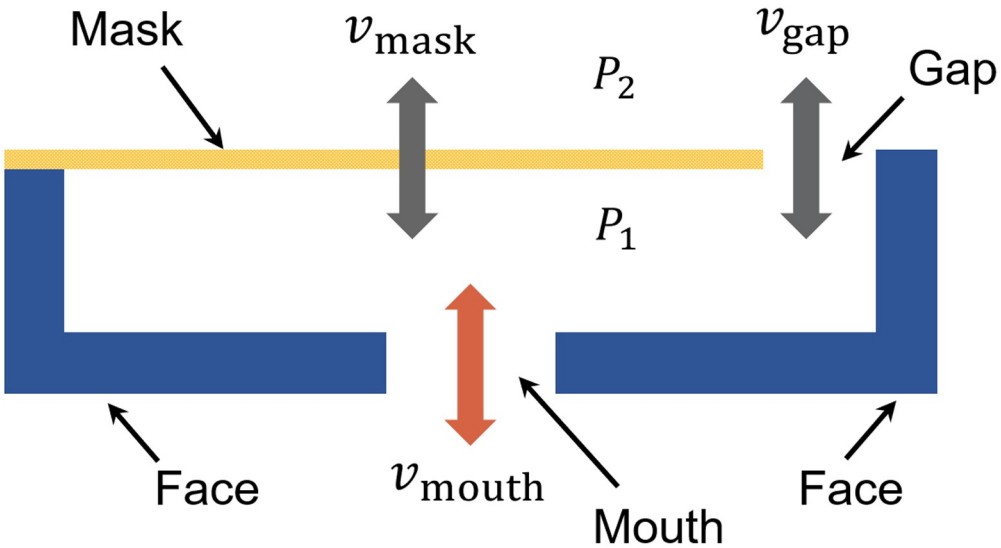

**Fig 2. The model of the mask with face-seal leakage.**

and the gap. The spatially-averaged mask and gap velocities are represented by $v_{\text{mask}}$ and $v_{\text{gap}}$, respectively. Similarly, $v_{\text{mouth}}$ denotes the average velocity of the air through the 'mouth' boundary.

Masks are generally composed of non-woven fabric outer layers and melt-blown fabric filter layers that can be considered (on a macroscopic level) as uniform porous layers, where fluid mechanics principles can be used to analyze the behavior of fluids flow through porous media. In examining flow, the magnitude of the Reynolds coefficient determines the appropriate mathematical formulation. In this study, $v_{\text{mask}} < 1 \text{ m·s}^{-1}$ (Calculated by Darcy' Law of Eq 5; for commercial masks, $\Delta P < 350$ Pa [29–31], and the ventilation resistance $\mu_{\text{air}} \cdot d_{\text{mask}} \cdot \kappa^{-1} \sim 1000 \text{ Pa·s·m}^{-1}$ [4]), air density $\rho_{\text{air}} \approx 1.29 \text{ kg·m}^{-3}$, the magnitude of the effective length ($L_{\text{eff}}$, i.e., mean hydraulic diameters of pores) is $O(10^{-5})$ [32], and the dynamic viscosity of the air, $\mu_{\text{air}} \approx 1.79 \cdot 10^{-5} \text{ kg·m}^{-1} \cdot \text{s}^{-1}$. Therefore, the Reynolds number of the mask can be expressed as

$$Re = \frac{\rho_{\text{air}} v_{\text{mask}} L_{\text{eff}}}{\mu_{\text{air}}}. \tag{4}$$

For $Re < 4$, Darcy's law can be applied to describe flow through porous material [33], with

$$-\frac{\Delta P}{d_{\text{mask}}} = \frac{\mu_{\text{air}}}{\kappa} v_{\text{mask}}, \tag{5}$$

where $\Delta P$ is the pressure drop across the mask thickness, $d_{\text{mask}}$ is the mask thickness and $\kappa$ is the value of Darcy's permeability for a given mask material (in m$^2$). Flow through the gap(s) can be described via Bernoulli's equation (excluding gravitation [34]), with

$$\frac{\Delta P}{\rho_{\text{air}}} + \frac{v_{\text{gap}}^2 - v_{\text{outside}}^2}{2} = 0. \tag{6}$$

Here $v_{\text{outside}}$ denotes the velocity of the air outside the mask, which is set to zero. Eq 6 therefore simplifies to

$$\frac{\Delta P}{\rho_{\text{air}}} + \frac{v_{\text{gap}}^{2}}{2} = 0. \tag{7}$$

We then combine Eqs 5 and 7 to obtain the relationship between $v_{\text{mask}_x}$ and $v_{\text{gap}_x}$, with

$$v_{\text{mask}_x} = \frac{\rho_{\text{air}}\kappa}{2d_{\text{mask}}\mu_{\text{air}}} v_{\text{gap}_x}^{2}, \tag{8}$$

which relates the velocity through the mask to the gap air velocity as a function of the mask and air properties. The subscript x can be replaced by *in* or *out* to indicate inhalation and exhalation, respectively. The total volumetric flow rates of inspiration and expiration are equal to the volumetric flow rate through the mask and the gap, so we rewrite Eq 1 to account for these separately, with

$$Q_x = \frac{\alpha_x \sin\,(\beta_x t)}{1000} = Q_{\text{mask}_x} + Q_{\text{gap}_x} = A_{\text{mask}} v_{\text{mask}_x} + A_{\text{gap}} v_{\text{gap}_x}, \tag{9}$$

Where the $Q_{\text{mask}_x}$ and $Q_{\text{gap}_x}$ are the flow rates through the mask and gap, respectively. $A_{\text{mask}}$ and $A_{\text{gap}}$ similarly are the surface area of mask and gap. Substituting Eq 8 into Eq 9 yields

$$Q_x = \frac{\alpha_x \sin\,(\beta_x t)}{1000} = A_{\text{mask}} \frac{\rho_{\text{air}}\kappa}{2d_{\text{mask}}\mu_{\text{air}}} v_{\text{gap}_x}^{2} + A_{\text{gap}} v_{\text{gap}_x}. \tag{10}$$

Eq 10 is a time-dependent quadratic equation. Therefore, $v_{\text{gap}_x}$ can be solved in terms of time:

$$v_{\text{gap}_x} = \frac{-A_{\text{gap}} \pm \sqrt{A_{\text{gap}}^{2} + 2Q_x \frac{A_{\text{mask}}\rho_{\text{air}}\kappa}{d_{\text{mask}}\mu_{\text{air}}}}}{\frac{A_{\text{mask}}\rho_{\text{gas}}\kappa}{d_{\text{mask}}\mu_{\text{gas}}}}. \tag{11}$$

It is worth noting that this equation has two solutions (due to the $\pm$ symbol), where only the solution with the same sign as $Q_x$ should be retained. The value of $v_{\text{mask}_x}$ can then be solved by substituting $v_{\text{gap}_x}$ into Eq 8. The time-dependent form of $v_{\text{gap}_x}$ (in Eq 11) and $v_{\text{mask}_x}$ (in Eq 8) is appropriate in the case of time-dependent inspiration/expiration flow rates, since this takes into account the sinusoidal nature of human breath. However, to obtain the time-independent expressions of $v_{\text{mask}_x}$ and $v_{\text{gap}_x}$ (i.e., with constant air flow), $Q_x$ in Eq 11 can be simply replaced with a constant breathing flow rate (in $\text{m}^3\cdot\text{s}^{-1}$).

## Filtration performance

In this study we define the filtration ratio, $\eta_x$, as the fraction of total airflow that passes through the mask material, with

$$\eta_x = \frac{V_{\text{mask}_x}}{V_{\text{tot}_x}} = 1 - \frac{V_{\text{gap}_x}}{V_{\text{tot}_x}}, \tag{12}$$

where the $V_{\text{tot}_x}$ is the total air flow through the mask and gap, equivalent to the respiratory in/outflow. $V_{\text{mask}_x}$ and $V_{\text{gap}_x}$ are the volume of air flow through the mask and the gap, respectively.

$V_{\text{gap}_x}$ is the integral of $v_{\text{gap}_x}$ and $A_{\text{gap}}$ over time, with

$$V_{\text{gap}_x} = A_{\text{gap}} \int_{t_0}^{t_1} v_{\text{gap}_x} \, dt. \tag{13}$$

From time $t_0$ to $t_1$, the value of $V_{\text{tot}_x}$ is given by:

$$V_{\text{tot}_x} = \int_{t_0}^{t_1} Q_x dt \tag{14}$$

Substituting Eqs 11 and 12 into Eq 10 and rewriting this in terms of the average efficiency between timepoints $t_a$ and $t_b$, this is expressed as

$$\eta_x = 1 - \frac{A_{\text{gap}} \int_{t_a}^{t_b} v_{\text{gap}_x} \, dt}{\int_{t_a}^{t_b} \dot{Q}_x dt}. \tag{15}$$

The overall total filtration ratio $\eta$ can then be found using the total ratio of mask vs. gap flow rates over one breath cycle, including both inspiration and expiration components, given by

$$\eta = 1 - \frac{|A_{\text{gap}} \int_{t_0}^{t_1} v_{\text{gap}_{in}} \, dt| + |A_{\text{gap}} \int_{t_1}^{t_2} v_{\text{gap}_{out}} \, dt|}{|\int_{t_0}^{t_1} Q_{in} dt| + |\int_{t_1}^{t_2} Q_{out} dt|}, \tag{16}$$

where $\eta$ is the filtration ratio over one breath cycle, with $[t_0, t_1]$ and $[t_1, t_2]$ denoting the inhalation and exhalation periods.

The mask width ($W_{\text{mask}}$) and the mask height ($H_{\text{mask}}$) were measured as $18.6 \pm 1.5$ cm and $16.7 \pm 1.3$ cm for two different types of P1/N95 masks and two different types of surgical masks. $A_{\text{mask}} = W_{\text{mask}} H_{\text{mask}}$ is the surface area of the mask.

The fit factor, $\eta_{\text{mask}}$, is distinct from the filtration ratio, where the mask material will only sort an (ideally high) fraction of the particulate and aerosol matter passing through the filter. The filtration ratio thus represents an upper bound for fit factor, with

$$\eta_{\text{mask}} = \frac{(1 - \eta_n) V_{\text{mask}_{in}} C_p + V_{\text{gap}_{in}} C_p}{(V_{\text{mask}_{in}} + V_{\text{gap}_{in}}) C_p} = \frac{(1 - \eta_n) V_{\text{mask}_{in}} + V_{\text{gap}_{in}}}{(V_{\text{mask}_{in}} + V_{\text{gap}_{in}})} \tag{17}$$

Where $\eta_n$ is the nominal (manufacturer-indicated) filtration efficiency mask rating (i.e. 0.95 for an N95 mask) and $C_p$ is the concentration of airborne particles (note that this cancels out in the right-hand side of the equation).

## Methods

### Analytical method

Analytical solutions were computed in MATLAB (R2019b, Mathworks Inc., Natick, MA). The equations and parameters in Eqs 1–3 and Table 1 are used to calculate the $Q_x$ vs. $t$ curves of a representative adult male and female [26–28]. These $Q_x$ vs. $t$ curves are then used in Eq 10 to solve the quadratic equation of $v_{\text{gap}_x}$ in terms of time, where the x subscript can be replaced by *in* or *out* to indicate inhalation and exhalation, respectively. The calculated $v_{\text{gap}_x}$ values are then substituted into Eq 16 to calculate the filtration ratio $\eta$. It is worth noting that for the different sex and breathing stages, the integration intervals $[t_0, t_1]$ (for inhalation) and $[t_1, t_2]$ (for exhalation) in Eq 16 are time variant, with the sign of $Q_x$ switching between exhalation and inhalation phases. Therefore, according to Eq 1, the integration interval $[t_a, t_b]$ in Eq 15 should be

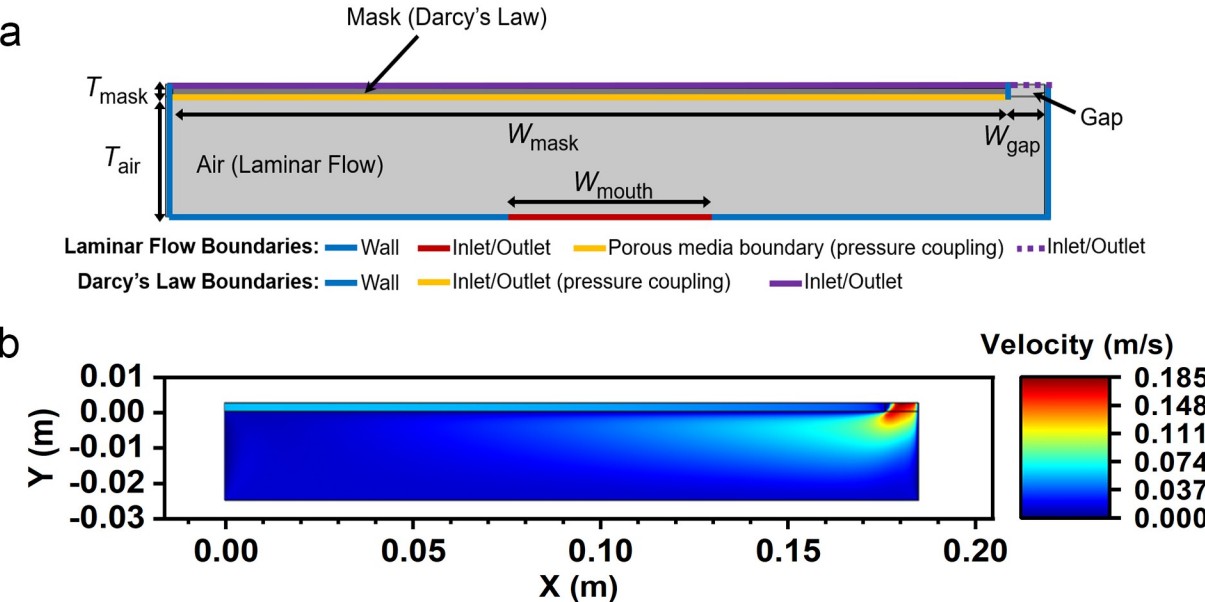

**Fig 3. Schematic diagrams of the model.** (a) Schematic diagram of FEM model and boundary conditions. Blue lines indicate solid walls, the red line indicates the inspiration/expiration location, the yellow line the inlet/outlet on the interior of the mask and the purple line the inlet/outlet on the mask exterior. (b) The velocity distribution during exhalation of a mask with $\sigma = 0.05$ and $R = 900$ Pa·s·m$^{-1}$.

half of the period (i.e. $t_b - t_a = \pi/\beta_x$) to represent the complete inhalation process, which also applies to the exhalation process.

## Finite element method

The FEM solution is computed in the COMSOL Multiphysics 5.5 environment with LiveLink™ for interfacing with a MATLAB interface (COMSOL Inc., MA, USA). Fig 3 shows the FEM model corresponding to the analytical boundary conditions in Fig 2, which couples the use of a mask material with defined permeability with the use of the time-dependent flow rates in a modeled geometry. Whereas the actual geometry in a worn physical mask will be highly dependent as a function of user and mask factors, we utilize this simplified model in order to provide generalized results that can provide useful relationships between gap size and resultant filtration ratio. As shown in Fig 3, the geometry of the model is designed in COMSOL Multi-physics according to the geometric dimensions of a representative mask, with $W_{mask} = 0.186$ m, and $W_{mouth} = 0.0463$ m [35] is the width of adults mouth and nostrils. For the purpose of this work, we use the word 'mouth' to refer to inspiration/expiration pathways, which in prac-tice include the mouth and nostrils. For the analytical model, $W_{mouth}$ is implicitly equivalent to the width of $W_{mask}$, since the velocities will be a function of a uniform pressure differential between the mask interior and exterior. Therefore, we also verify the case of $W_{mouth} = W_{mask}$ in the FEM method. Whereas these dimensions have discrete units in the simulation model, we normalize these with $W_{gap}$, denoted as $W_{gap} = \sigma W_{mask}$, where $\sigma$ is ratio of the gap over the mask dimensions, in order to yield a useful and generalizable measure of $\eta$. We set the mask thickness to a discrete representative value, $T_{mask} = 2.51$ mm [4], where we vary instead the overall permeability of the mask material $\kappa$ as an independent parameter. For simplicity, a rep-resentative distance between the mask and the mouth, $T_{air}$, is represented by $T_{air} = 10T_{mask}$.

As shown in Fig 3a, Darcy's Law and the Laminar Flow physics module are applied to the mask domain (darker gray) and air domain (light gray), respectively. Fig 3a describes the geometric elements corresponding to each boundary condition. The coupling between the two physical modules is achieved by defining the pressure coupling at the boundary between the Laminar Flow module and Darcy's Law module (shown as the yellow boundary in Fig 3a). Therefore, the pressure of the Laminar Flow module at the wall is the same as the inlet pressure boundary condition of Darcy's Law module. For the inlets/outlets represented by purple boundaries, the boundary conditions are defined as pressure equal to zero to represent the atmospheric zero (gauge) pressure outside the mask; the dashed purple line in Fig 3a represents the inlet/outlet boundary adjoining the gap region, whereas the solid purple line represents the inlet/outlet adjoining the mask domain. We make this boundary distinct from the yellow line on the interior of the mask as no pressure coupling is used on the mask exterior, given that the latter is defined as having a zero (gauge) pressure. The red boundary in Fig 3a represents the mouth and nostrils, which is the inlet/outlet boundary. The velocity boundary condition $v_{\mathrm{mouth}_x}$ is defined as $Q_x/A_{\mathrm{mouth}}$. The materials of the mask and air domain are set to Porous Matrix and Air, respectively. The air material is defined by the built-in properties of COMSOL Multiphysics ($\rho_{\mathrm{air}} = 1.204$ kg·m³ and $\mu_{\mathrm{air}} = 1.0884$ Pa·s, at 20˚C). The porosity in the porous matrix properties is set to 0.9 [36], and the permeability $\kappa$ is defined according to the different mask materials examined in this study. For Darcy's law in Eq 5, $v_{\mathrm{mask}}$ is a function of permeability, $\kappa$, as well as $d_{\mathrm{mask}}$. To make the study applicable to masks with various permeability and thickness, we introduce the mask ventilation resistance term, $R$, which is inversely proportional to the permeability and incorporates the thickness of mask material, with

$$R = \frac{\mu_{\mathrm{air}}}{\kappa} d_{\mathrm{mask}}, \tag{18}$$

with units of Pa·s·m⁻¹. The range of $R$ in this study is set to 500–2500 Pa·s·m⁻¹, representative of the range of permeability values in typical mask materials [37].

To study masks with different $R$, first the corresponding $\kappa$ are calculated for various $R$ according to Eq 17. Then each $\kappa$ is input into the COMSOL numerical model as the permeability property of the porous matrix. The value of σ then varies from 0 to 0.05 (with the step size of 0.002) to represent normalized gap areas between 0 and 5% of the mask area. The mesh of the model is refined to a maximum $2 \times 10^{-4}$ m to a minimum $\approx 9.1 \times 10^{-5}$ m to ensure the mesh is dense enough for the smallest modelled gap area (σ = 0.002, that is, $W_{\mathrm{gap}} \approx 4 \times 10^{-4}$ m), and where deviations in this mesh size parameter do not alter the filtration ratio below $2 \times 10^{-4}$ m. The mesh distribution of σ = 0.05 is shown in S1 Fig. The time-dependent solver is used to solve the model. After the integral calculation according to Eq 16, MATLAB code is used to process the exported data. Fig 3b shows a representative velocity distribution with σ = 0.05 and $R$ = 900 Pa·s·m⁻¹, where the highest velocities occur in the vicinity of the gap at the right side in Fig 3b. The simulation model here is comprised of a mask geometry with a single-sided gap. The pressure distribution of the single-sided gap model along the length of the mask is more uneven than the masks with the gaps on two sides, especially in the case of a modelled condition with a finite-width mouth, resulting in a deviation from the analytical model when the gap increases (see S2 Fig). On the contrary, a far smaller difference (mean value <5%) is observed between the double-sided gap model and the analytical model, which is attributed to the more uniform pressure distribution over the internal mask interface. We nevertheless show results for a single-sided gap our figures in order to highlight the maximum deviation possible from our analytical model in the case of a finite width mouth serving as the inlet/outlet boundary condition inside the mask.

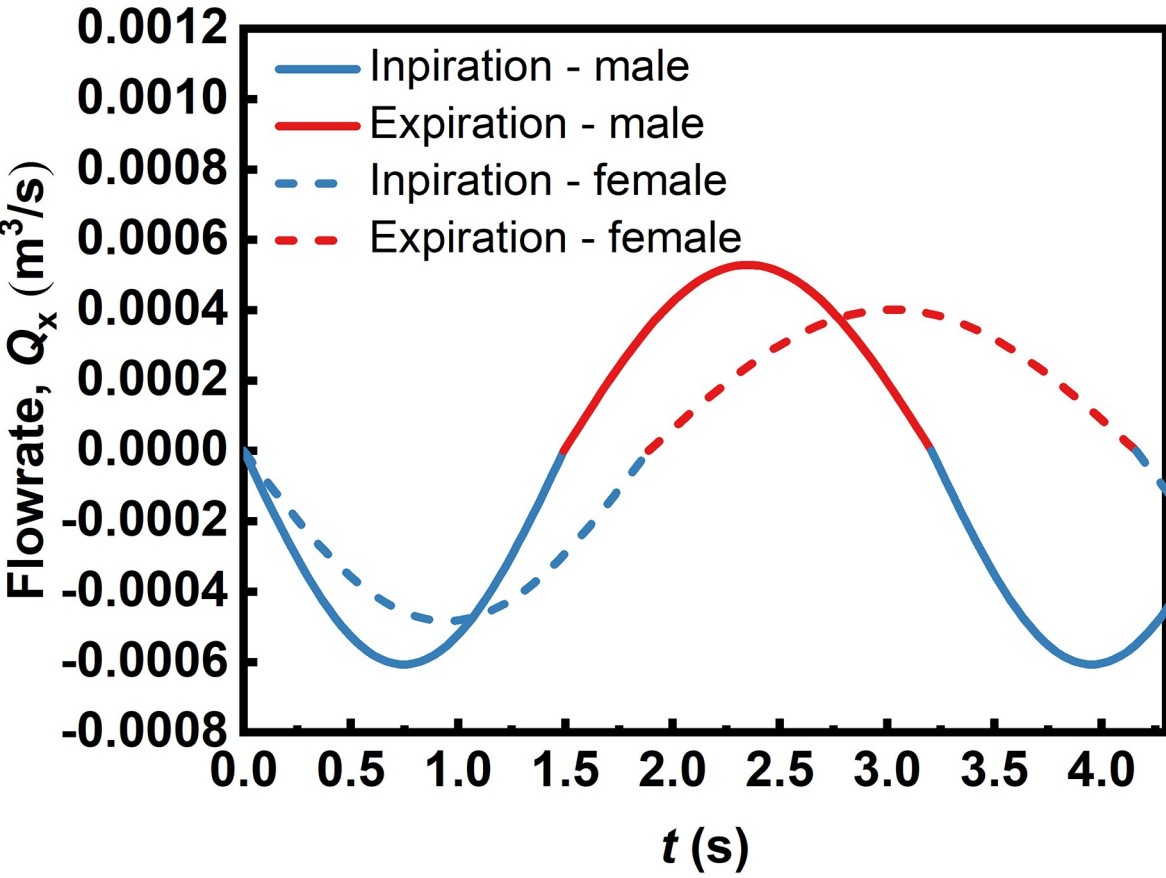

**Fig 4. Representative inspiration/exhalation flowrate vs. time curves for a representative adult male and female obtained using the model in the respiration model section.** The coefficients that describe these curves as per Eq 1, namely $\alpha_{in}$, $\beta_{in}$, $\alpha_{out}$, and $\beta_{out}$, are 0.61, 2.10, 0.53, 1.83 for males and 0.48, 1.66, 0.40, 1.38 for females, respectively.

## Results and discussion

### Respiration cycle

Fig 4 shows the representative breathing curves of an adult male and female. The inhalation and exhalation period are represented by negative and positive $Q_{in}$ values, respectively. As shown in Fig 4, the inhalation period exhibits a higher peak flow rate than exhalation, with a shorter inhalation period than for exhalation. The peak flow rates for males are $-6.068 \cdot 10^{-4}$ $m^3 \cdot s^{-1}$ (inhalation) and $5.284 \cdot 10^{-4}$ $m^3 \cdot s^{-1}$ (exhalation), and for females are $-4.836 \cdot 10^{-4}$ $m^3 \cdot s^{-1}$ (inhalation) and $4.014 \cdot 10^{-4}$ $m^3 \cdot s^{-1}$ (exhalation). The modeled inhalation and expiration periods are 1.49 s and 1.71 s for males, and 1.89 s and 2.27 s for females, respectively.

### The impact of the face-seal leakage

The contours of efficiency $\eta$ as a function of mask resistivity and gap size ratio ($R$ and $\sigma$) using both analytical method (Eq 16) and FEM simulations are shown in Fig 5, where $\eta$ is the ratio of the air flow through the mask to the total air flow (the total air flow through the mask and the gap). Fig 5a shows the analytical results for a representative male and female, respectively. Fig 5b shows the corresponding simulation results with $W_{mouth} = W_{mask} + W_{gap}$. As opposed

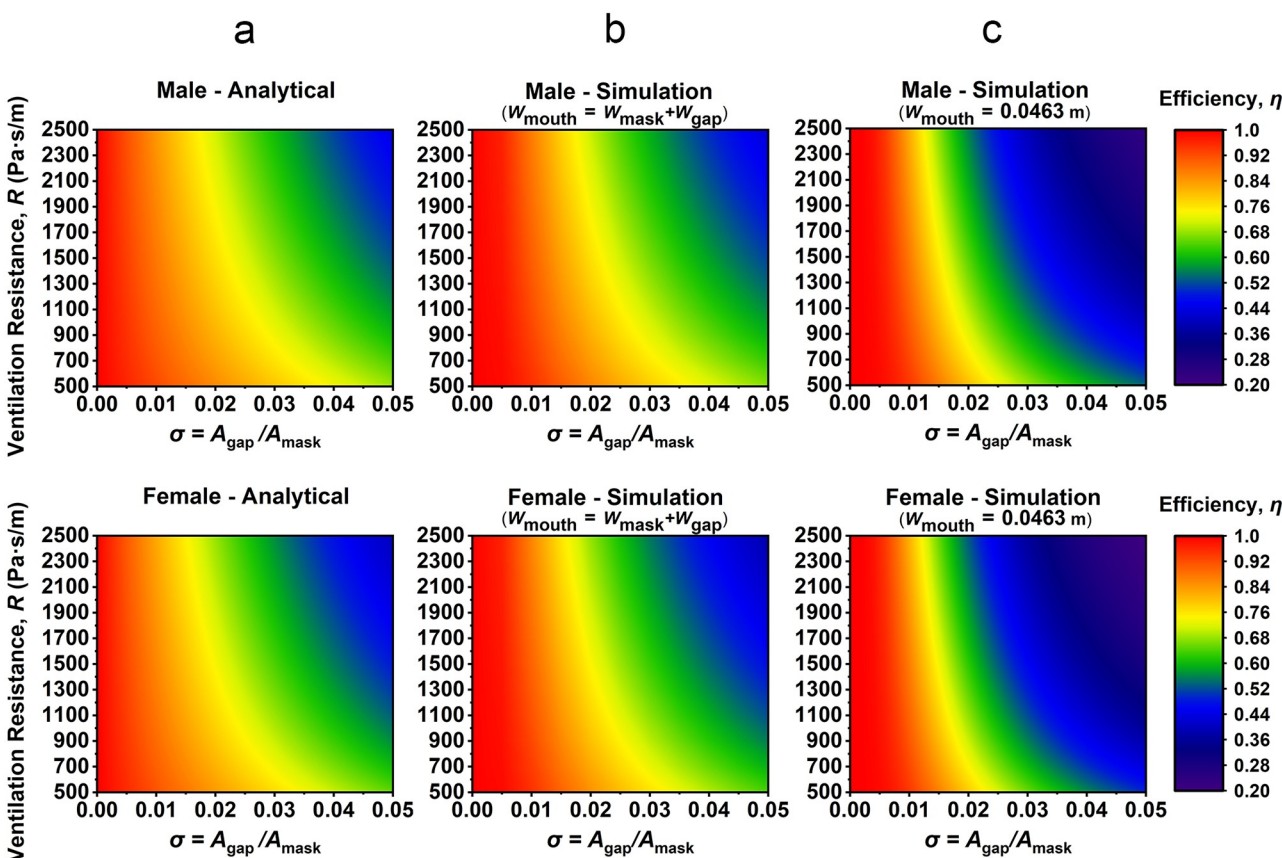

**Fig 5. The relationships between σ, R, and η of different sex.** (a) Analytical result, (b) simulation result with mouth inlet/outlet extending the width of the simulation domain ($W_{mouth} = W_{mask} + W_{gap}$), and (c) simulation result with a finite mouth width ($W_{mouth} = 0.0463$ m).

to Fig 5b in which the airflow inlet/outlet boundary condition inside the mask is uniform across the modelled domain (bottom wall in Fig 3a), Fig 5c shows the simulation results in which we incorporate a physical limitation on the mouth dimensions in our simulation model, with $W_{mouth} = 0.0463$ m. Each result did not show significant difference in η between male and female (mean difference < 3%). As shown in Fig 5, the trend is that η decreases for a given R with increasing gap size σ, where the degradation in η vs. σ occurs at a higher rate for higher R values. This indicates that a larger fraction of the airflow is directed through the mask gaps with higher R values, especially true when σ is larger. This phenomenon is evident in Eq 8, where κ (permeability) and R are inversely correlated, and where a decrease in κ results in a lower airflow velocity through the mask. Further, as per Eq 11, $v_{gap_x} \propto \kappa^{-0.5}$, which means that a decrease of κ will simultaneously result in the increase of gap airflow velocity.

Comparing Fig 5a and 5b, we can observe no significant difference between the analytical results and the simulation results (mean difference < 1.5%), demonstrating the validity of the analytical approach in modelling a simplified system. Since the analytical model in Eq 16, however, is a generalized equation that is not specific to a particular geometry, we seek to understand how a spatially limited inlet/outlet condition (i.e. a mouth) might impact relative airflow pathways through masks vs. gap. Interestingly, this simulation approach results in a marginally higher efficiency for small gap dimensions, but significantly lower efficiency for much larger gap dimensions. Nevertheless, a quantitative examination of Fig 5c show that when the inlet/

outlet condition is set to a value representative of a mouth width ($W_{\text{mouth}}$ = 0.0463 m), there is only a small difference from prior analytical/simulation results when the face-seal gap is small (i.e. $\sigma < 0.015$, mean difference < 3%), with the most significant deviation for $\sigma > 0.02$ (mean difference > 5%). Limiting our analysis to an intermediate gap size range with $0.015 < \sigma < 0.02$ for values of $R < 1500$ Pa·s·m$^{-1}$, which is the case for the example masks resistivities given later in this work, there is also still good agreement with the analytical results (mean difference < 3%). This deviation at higher $\sigma$ values is ultimately caused the concentration of airflow in the middle of the mask, with a resulting uneven pressure distribution across the mask interior (see S3 Fig). Therefore, the pressure is unevenly distributed along the coupling boundary between laminar flow and Darcy's law (i.e., the yellow boundary in Fig 3a). However, in the analytical model, the coupling boundary is idealized as equal pressure along the boundary. In the case of a fixed volume flow rate, $v_{\text{mouth}}$ increases as the mouth size decreases. The increased $v_{\text{mouth}}$ result in the increased local pressure near the mask and gap interior (see S3 Fig). As per Eq 8, the increased local pressure will increase the velocity through gap and mask, whereas the change in $v_{\text{mask}_x}$ is greater because $v_{\text{mask}_x} \propto v_{\text{gap}_x}^2$ while $v_{\text{gap}_x} < 1$ m·s$^{-1}$. For all cases however, we find that the ventilation resistance has surprisingly little effect on filtration ratio for small gap dimensions ($\sigma < 0.015$).

Fig 6 shows the dependences between $\eta$ and $\sigma$ for representative 3M 1860 and 1870+ respirator masks, with common use in medical environments [38–40]. Here the $R$ for 3M 1860 is 928 Pa·s·m$^{-1}$, and $R = 1272$ Pa·s·m$^{-1}$ for the 3M 1870+ [4]. The solid lines in Fig 6a and 6b correspond to the results in Fig 5a, the dotted line corresponds to Fig 5b and the dashed line corresponds to Fig 5c for these discrete $R$ values. These figures show that the analytical method and the simulation methods display similar behavior over the range of $\sigma$ shown, with decreasing efficiency with increasing mask gap dimensions, where insets show minimal (maximum ~5%) deviation for $\sigma < 0.015$ between analytical models and both simulation setups, with the analytical model giving more conservative efficiency reduction predictions. For larger gap sizes the analytical and simulation case (with uniform flow across the inlet/outlet corresponding to the implicit assumptions in the analytical model, dotted lines) results are essentially identical. As in Fig 5, incorporating a finite-width inlet/outlet condition (dashed lines) shunts additional airflow through the gap for intermediate and larger gap values ($\sigma > 0.015$).

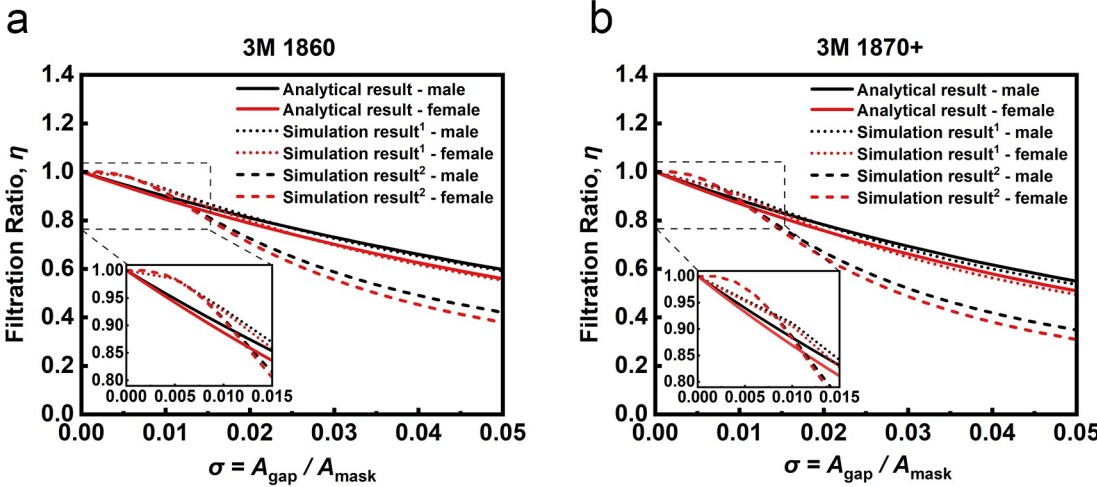

**Fig 6. The dependences between $\eta$ and $\sigma$ for (a) 3M 1860 and (b) 1870+ masks.** Dotted lines, 1, with $W_{\text{mouth}} = W_{\text{mask}} + W_{\text{gap}}$. Dashed lines, 2, showing the case a finite modelled mouth width $W_{\text{mouth}} = 0.00463$ m.

Interestingly, there is minimal difference between the performance of masks for male and female wearers for small gap sizes, with only marginally worse filtration ratio for the representative female inspiration model, though this difference is amplified for larger gap sizes. This can be explained by the lower average flow rate for our representative female subject compared to the male (as shown in Fig 4), leading to an increase in the ratio of $V_{gap_x}$ to $V_{tot_x}$ according to Eqs 10 and 16 and thus a smaller value of $\eta$, where more air is directed through the mask with a greater pressure differential between the interior and exterior of the mask. There is minimal overall difference between the performance of these two example masks or the trends in the modeled responses, though the filtration ratio for the higher resistivity mask is lower for all simulation and analytical models across all σ values.

In Fig 7 we seek to estimate the impact of the mask fit on the fit factor, the fraction of the airborne particles in the design size range that are captured by the mask. In this analysis we assume that particles going through the gap are not filtered, whereas the airflow going through the mask has a percentage of its particle load filtered according to the mask specifications (i.e. an N95 mask filters > 95% of 0.3 μm particles [41]). We use the simulation model from Fig 5c here, since this is the most representative of the boundary conditions of the two simulation cases for a physical mask. Fig 7a shows the fit factor of airborne particles calculated via Eq 17, shows results for 1860 and 1870+ masks at $0 < \sigma < 0.016$. We show results in this range (as opposed to $0 < \sigma < 0.05$) to limit our analysis to the range of the common mask classifications [3–6]. The black dashed lines in the figure are the particle filtration rate corresponding to different mask classifications [42, 43], where sufficiently large gap dimensions will reduce a given N95 mask's equivalent performance to these lower standards. The error bars here reflect the N95 standard that > 95% of particles are captured in a perfectly sealed mask (i.e. a filter material with a 100% capture rate would also conform to this standard). Notably, when σ is negligible (σ < 0.005, less than 0.5% gap size), the effect of gap on the filtration rate is minimal with reduction in filtration ratio of < 2.1%. The reduction in fit factor for larger gap sizes, however, is steep, with the masks capturing at most 80% of particulates for gap ratios greater than ~1.5% (lower than the FFP1 standard), regardless of the initial filtration efficiency of the mask.

Whereas Fig 7a shows filtration efficiencies for specific mask filter materials, Fig 7b shows the filtration efficiencies for a much wider range of ventilation resistance values for an N95

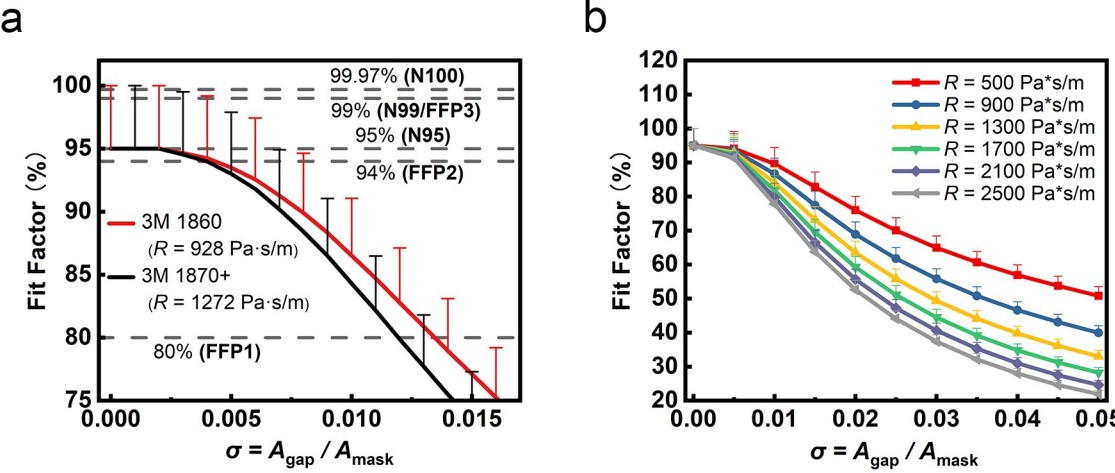

**Fig 7. The results of gap area vs. fit factor.** For (a) representative commercially available masks (3M 1860 and 1870+) and (b) N95 masks with various ventilation resistances.

mask, from 500 to 2500 Pa·s·m$^{-1}$. Regardless, the curves for these different resistances all show similar trends, with decreasing fit factor for larger values of σ. When $R$ = 500 Pa·s·m$^{-1}$, the fit factor drops from 95% to 50.81% between 0 < σ < 0.05. However, in line with prior results showing decreased performance for larger resistance, for $R$ = 2500 Pa·s·m$^{-1}$ the filtration ratio is reduced from 95% to 21.9%. This is consistent with the prediction in Eq 11 that the reduced permeability results in an increased $v_{gap_x}$ Also of note is the reduction in the decreasing error bar with increasing σ, since its magnitude scales with the fraction of airflow passing through the mask material.

## Conclusion

In this article we quantified the impact of mask seal gaps on filtration ratio and fit factor via analytical and simulation approaches, with application to mask materials with a wide range of ventilation resistances and mask gap areas. Our results show that the face-seal leakage has a significant impact on the fraction of airflow passing around the filter material, where both increased ventilation resistance and mask gap dimensions degrade mask performance. For mask areas on the order of ~200–300 cm$^2$, for example, gap dimensions corresponding to just 1.5% of the mask area (~4 cm$^2$) can result in approximately 20% of the airflow bypassing the filter for typical materials. This unfiltered 20% has an outsized impact on performance; since the relationship between infection risk and viral particle load is non-linear, 80% of air being filtered equates to much less than an 80% infection risk reduction [44].

Compared with previously reported methods that were specific to particular mask geometries and experimental protocols, this analytical and FEM evaluation can be applied universally to readily provide first-order estimates of mask performance, and therefore permissible gap dimensions. Moreover, this approach is equally applicable to full face respirators, surgical masks and respiratory protection masks. Our results highlight the critical importance of not only reducing gap dimensions, where gaps of greater than 1% can result in significant fractions of airflow bypassing the filter material, but also of maximally reducing ventilation resistance (though not at the expense of particulate filtering performance). Choosing engineered nanofiber materials as opposed to the more typical melt-blown filter layer [3, 4, 45, 46], for example, may help to make mask performance less sensitive to small gaps. Interestingly, we also find that very small gap sizes (less than 0.5% of the mask area) are likely to have a minimal impact on fit factor, potentially allowing future design to strike a balance between comfort and fit factor in select usage scenarios. In addition, the simulation and analytical models developed here have the potential to assess the impact of usage-associated changes in filter characteristics, where the accumulation of particles and moisture can alter the filter's flow resistivity. Our results show that an increase in resistivity, for example, results in a decrease in the fraction of air that passes through the filter material, such that the performance of a pristine filter is likely to deteriorate with use in the presence of mask gaps. Whereas the analytical model demonstrates the scaling relationship between bypass ratio, gap dimensions and mask materials for an idealized mask setup, the simulation modelling approach might also be further utilized to examine the impact of non-uniform accumulation of particulate matter as well via the alteration of the localized flow resistivity parameters.

## Supporting information

**S1 Fig. The mesh distribution of σ = 0.05.**
(TIF)

**S2 Fig. Filtration ratios of single-sided gap, double-sided gaps, and analytical result.** For (a) 3M 1860 and (b) 3M 1870+ masks.
(TIF)

**S3 Fig. Representative pressure distribution diagram.** For (a) $W_{\mathrm{mouth}} = W_{\mathrm{mask}} + W_{\mathrm{gap}}$ and (b) $W_{\mathrm{mouth}} = 0.0463$ m. With $\sigma = W_{\mathrm{mask}}/W_{\mathrm{gap}} = 0.03$ and $Q_{\mathrm{tot}} = 0.00035$ m$^3$·s$^{-1}$.
(TIF)

**S1 Data.**
(XLSX)

# Author Contributions

**Conceptualization:** Peter Lee, David Collins.

**Formal analysis:** Mingxin Xu.

**Investigation:** Mingxin Xu.

**Methodology:** Mingxin Xu.

**Supervision:** Peter Lee, David Collins.

**Writing – original draft:** Mingxin Xu.

**Writing – review & editing:** Peter Lee, David Collins.

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
