## [Decision Letter · Decision Letter 0]

9 Dec 2020

PONE-D-20-32100

The critical importance of mask seals on respirator performance: an analytical and simulation approach

PLOS ONE

Dear Dr. Collins,

Thank you for submitting your manuscript to PLOS ONE. After careful consideration, we feel that it has merit but does not fully meet PLOS ONE’s publication criteria as it currently stands. Therefore, we invite you to submit a revised version of the manuscript that addresses the points raised during the review process.

We look forward to receiving your revised manuscript.

Kind regards,

Dahua Shou

Academic Editor

PLOS ONE

Journal Requirements:

2.We note that Figure [1] includes an image of a patient / participant in the study. 

Reviewers' comments:

Reviewer's Responses to Questions

**Comments to the Author**

1. Is the manuscript technically sound, and do the data support the conclusions?

Reviewer #1: Yes

Reviewer #2: Yes

2. Has the statistical analysis been performed appropriately and rigorously? 

Reviewer #1: Yes

Reviewer #2: I Don't Know

3. Have the authors made all data underlying the findings in their manuscript fully available?

Reviewer #1: Yes

Reviewer #2: Yes

4. Is the manuscript presented in an intelligible fashion and written in standard English?

Reviewer #1: Yes

Reviewer #2: Yes

5. Review Comments to the Author

Reviewer #1: This study investigates the effect of mask sealing, which is significant but has less understanding in literature. The author calculates the influence of face-seal leakage on the filtration efficiency based on FEM model and analytical model. Good agreement between analytical and numerical models are obtained. This study points out an importance conclusion on the significance of forming a tight seal of mask. The language and results are well organized. However, the following issues should be addressed before accepting for publication:

1. The author should double check the format issue. For example, there should be a dot after the abbreviation. For example, the “Fig 7a” should be “Fig. 7a” in line 390.

2. In a real mask, the inner space is 3D curved shape, with gaps in both sizes. However, in this study, the region is simplified to be a rectangular, with gap only in one side. Can the results of the simplified case applied to that of the real case? The author should further state the difference between these two situations.

3. In Fig.3 (a), the yellow line is defined as a “wall”. However, the definition of a “wall” in CFD indicate the boundary with no permeability. The phrase should be modified to be “porous media” or something more appropriate.

4. In Fig.3 (a), what is the solid purple line indicated? It seems unnecessary.

5. In the caption of Fig.3 (b), line 245, is the velocity distribution generated in inspiration or expiration process? It should be stated clearly.

6. In the line 275, the author mentioned that the refined mesh size is 0.2 mm. However, more detail should be provided to state that the result has been independent with the mesh size.

Reviewer #2: This manuscript describes the application of an analytical model and finite element model to determine the amount of air that bypasses masks or respirators with poor fit during inhalation and exhalation. The authors perform the analysis in order to demonstrate the importance of a tight seal between face and mask/respirator to mask effectiveness. The models appear to be sound and produce interesting results in terms of the fluid flow through masks and around edges for simplified geometries. The findings that fit are important for filtration efficiency, however, are not surprising. The importance of a tight seal (good mask fit) has been very well understood for a long time. In fact, fit testing of respirators is mandated in the United States by the Occupational Safety and Health Administration if they are required in the workplace.

Specific comments:

Filtration efficiency has a very specific definition when applied to respirators and masks – related to the penetration by 300 nm particles under specific pressure and flow conditions over a specific unit area. Here the term appears to be redefined to describe the fraction of particles that passes around and through a mask. To avoid confusion with the more standard term, use of an alternative term is recommended. Fit factor, may be a more appropriate concept.

Lines 406-410: These results are in general agreement with published studies on the importance of fit that use physical measurements rather than model simulations. A more important application of the authors’ model might be in assessing the performance over time of respirator materials as moisture and particles build up in specific areas of the mask/respirator where flows are most concentrated.

6. PLOS authors have the option to publish the peer review history of their article (what does this mean?). If published, this will include your full peer review and any attached files.

Reviewer #1: **Yes: **Guanghan Huang

Reviewer #2: No

---

## [Author Response · Author response to Decision Letter 0]

21 Jan 2021

Dear reviewers,

Thank you for your valuable comments. We have made a point-by-point response to all the reviewers' comments in the following text. 

Reviewer #1:

This study investigates the effect of mask sealing, which is significant but has less understanding in literature. The author calculates the influence of face-seal leakage on the filtration efficiency based on FEM model and analytical model. Good agreement between analytical and numerical models are obtained. This study points out an importance conclusion on the significance of forming a tight seal of mask. The language and results are well organized. However, the following issues should be addressed before accepting for publication:

Response:

We appreciate the time and effort in providing insightful comments on our manuscript. We have made substantial modifications to both the text and figures as a result as detailed below.

1. The author should double check the format issue. For example, there should be a dot after the abbreviation. For example, the “Fig 7a” should be “Fig. 7a” in line 390.

Response:

Thank you very much for pointing this out. Based on the PLOS ONE format requirements (cite figures as “Fig 1”, “Fig 2”, etc.), we have checked and revised the format of this article and made this consistent throughout. 

2. In a real mask, the inner space is 3D curved shape, with gaps in both sizes. However, in this study, the region is simplified to be a rectangular, with gap only in one side. Can the results of the simplified case applied to that of the real case? The author should further state the difference between these two situations.

Response:

Thank you for this comment. We agree that the geometry will have some impact on the results, however there are a multitude (essentially the number of mask wearers times the number of different mask varieties) of non-uniform geometries that will be formed by the interface between the face and the mask. Even if a different geometry were used, we would still be simulating a particular mask shape (and implicitly a given face shape as well). Rather than attempt to model a specific combination of mask and face, the purpose of this manuscript is to delineate the scaling relationships according to both analytical and simulation models that show the effect on mask bypass ratios as a function of fit parameters (whether there are gaps). The specific geometry in non-inertially dominated flow, however, is unlikely to have a as large as an impact as other parameters. The position(s) of the gaps will also have an impact, which we believe is worthy of analysis. Accordingly, we have added a description of the effect of unilateral/bilateral gaps and gap geometry on the results:

“The simulation model here is comprised of a mask geometry with a single-sided gap. The pressure distribution of the single-sided gap model along the length of the mask is more uneven than the masks with the gaps on two sides, especially in the case of a modelled condition with a finite-width mouth, resulting in a deviation from the analytical model when the gap increases (see Supporting information, S2 Fig). On the contrary, a far smaller difference (mean value <5%) is observed between the double-sided gap model and the analytical model, which is attributed to the more uniform pressure distribution over the internal mask interface. We nevertheless show results for a single-sided gap our figures in order to highlight the maximum deviation possible from our analytical model in the case of a finite width mouth serving as the inlet/outlet boundary condition inside the mask.”

3. In Fig.3 (a), the yellow line is defined as a “wall”. However, the definition of a “wall” in CFD indicate the boundary with no permeability. The phrase should be modified to be “porous media” or something more appropriate.

Response:

Thank you for pointing this out. We have modified the corresponding legend to "Porous media boundary" in Fig. 3.

4. In Fig.3 (a), what is the solid purple line indicated? It seems unnecessary.

Response:

Thank you for this suggestion. The purple solid line is defined as the inlet/outlet of Darcy's law module in COMSOL. We use the solid line and the dashed purple lines to indicate the inlet/outlet boundary of the Darcy's law and the laminar flow module, respectively. Since the line is defined as inlet/outlet in COMSOL, we retain the purple solid line. 

“For the inlets/outlets represented by purple boundaries, the boundary conditions are defined as pressure equal to zero to represent the atmospheric zero (gauge) pressure outside the mask; the dashed purple line in Fig 3a represents the inlet/outlet boundary adjoining the gap region, whereas the solid purple line represents the inlet/outlet adjoining the mask domain. We make this boundary distinct from the yellow line on the interior of the mask as no pressure coupling is used on the mask exterior, given that the latter is defined as having a zero (gauge) pressure.”

5. In the caption of Fig.3 (b), line 245, is the velocity distribution generated in inspiration or expiration process? It should be stated clearly.

Response:

Thank you for pointing this out. We have modified the description as:

“The velocity distribution during exhalation of a mask with σ = 0.05 and R = 900 "Pa∙s∙" "m" ^"-1" .”

6. In the line 275, the author mentioned that the refined mesh size is 0.2 mm. However, more detail should be provided to state that the result has been independent with the mesh size.

Response:

Thank you for this comment. The choice of mesh dimensions should be such that the filtration ratio results are independent of mesh element size; deviations above and below a chosen mesh size should result in no (or very little) change in results. To assess the effect of mesh size on we examined a gap ratio given that demonstrated the most significant sensitivity to input parameters (i.e. different flow rates w/ male vs female modelled inputs). We used σ=0.05 (that is, A_gap /A_mask =0.05), which also has the maximum deviation from the theoretical model in the target σ range and verified the filtration ratio of 3M 1860 mask with the maximum mesh size ranges from 0.0001 m (half of 0.0002 m) to 0.002 m (ten times of 0.0002 m). The result shows that the maximum mesh size = 0.0002 m is within the convergence interval. 

Maximum mesh size (m) Filtration Ratio, η

0.00010 0.38314

0.00015 0.39127

0.00020 0.38993

0.00040 0.38972

0.00060 0.39036

0.00080 0.38997

0.00100 0.38959

0.00120 0.38921

0.00140 0.38883

0.00160 0.38880

In addition, we have modified the description of the mesh size as following:

“The mesh of the model is refined to a maximum "2 × " 〖"10" 〗^"-4" m to a minimum ≈"9.1 × " 〖"10" 〗^"-5" m to ensure the mesh is dense enough for the smallest modelled gap area ("σ = 0.002" , that is, W_gap≈"4 × " 〖"10" 〗^"-4" m), and where deviations in this mesh size parameter do not alter the filtration ratio below "2 × " 〖"10" 〗^"-4" m. The mesh distribution of "σ = 0.05" is shown in Supporting information, Fig. S1”

Reviewer #2:

This manuscript describes the application of an analytical model and finite element model to determine the amount of air that bypasses masks or respirators with poor fit during inhalation and exhalation. The authors perform the analysis in order to demonstrate the importance of a tight seal between face and mask/respirator to mask effectiveness. The models appear to be sound and produce interesting results in terms of the fluid flow through masks and around edges for simplified geometries. The findings that fit are important for filtration efficiency, however, are not surprising. The importance of a tight seal (good mask fit) has been very well understood for a long time. In fact, fit testing of respirators is mandated in the United States by the Occupational Safety and Health Administration if they are required in the workplace.

Response:

Thank you for your valuable feedback on our manuscript. We agree that fit testing is vitally important, where this manuscript further seeks to highlight its importance by examining the degradation in performance in the case of even small gap dimensions. This is important for not only clinical settings in which individual fit testing occurs, but for evaluating the epidemiological impact of mask wearing in the wider public (where this does not occur). In doing so we develop a straightforward analytical model that can predict this behavior. We have incorporated your suggestions throughout the manuscript as detailed below.

1. Filtration efficiency has a very specific definition when applied to respirators and masks – related to the penetration by 300 nm particles under specific pressure and flow conditions over a specific unit area. Here the term appears to be redefined to describe the fraction of particles that passes around and through a mask. To avoid confusion with the more standard term, use of an alternative term is recommended. Fit factor, may be a more appropriate concept.

Response:

Thank you for this suggestion. We agree with this and have changed the corresponding terms "filtration efficiency" to "fit factor".

2. Lines 406-410: These results are in general agreement with published studies on the importance of fit that use physical measurements rather than model simulations. A more important application of the authors’ model might be in assessing the performance over time of respirator materials as moisture and particles build up in specific areas of the mask/respirator where flows are most concentrated.

Response:

Thank you for this suggestion. Accordingly, we have added the following text to the corresponding paragraph to emphasize this potentially important application.

“In addition, the simulation and analytical models developed here have the potential to assess the impact of usage-associated changes in filter characteristics, where the accumulation of particles and moisture can alter the filter’s flow resistivity. Our results show that an increase in resistivity, for example, results in a decrease in the fraction of air that passes through the filter material, such that the performance of a pristine filter is likely to deteriorate with use in the presence of mask gaps. Whereas the analytical model demonstrates the scaling relationship between bypass ratio, gap dimensions and mask materials for an idealized mask setup, the simulation modelling approach might also be further utilized to examine the impact of non-uniform accumulation of particulate matter as well via the alteration of the localized flow resistivity parameters.”

---

## [Editor Report · Decision Letter 1]

26 Jan 2021

The critical importance of mask seals on respirator performance: an analytical and simulation approach

PONE-D-20-32100R1

Dear Dr. Collins,

We’re pleased to inform you that your manuscript has been judged scientifically suitable for publication and will be formally accepted for publication once it meets all outstanding technical requirements.

Kind regards,

Dahua Shou

Academic Editor

PLOS ONE
---

## [Editor Report · Acceptance letter]

5 Feb 2021

PONE-D-20-32100R1 

The critical importance of mask seals on respirator performance: an analytical and simulation approach 

Dear Dr. Collins:

I'm pleased to inform you that your manuscript has been deemed suitable for publication in PLOS ONE. Congratulations! Your manuscript is now with our production department. 

Kind regards, 

on behalf of

Dr. Dahua Shou 

Academic Editor

PLOS ONE